# Underwater Hyperspectral Target Detection with Band Selection

**Xianping Fu [1,2], Xiaodi Shang [1], Xudong Sun [1,2], Haoyang Yu [1], Meiping Song [1,*] and Chein-I Chang [1,3,4]**

[1] Information Science and Technology College, Dalian Maritime University, Dalian 116026, China; fxp@dlmu.edu.cn (X.F.); shangxd329@dlmu.edu.cn (X.S.); sxd@dlmu.edu.cn (X.S.); yuhy@dlmu.edu.cn (H.Y.); cchang@umbc.edu (C.-I.C.)
[2] Peng Cheng Laboratory, Shengzhen 518000, China
[3] Remote Sensing Signal and Image Processing Laboratory, Department of Computer Science and Electrical Engineering, University of Maryland, Baltimore, MD 21250, USA
[4] Department of Computer Science and Information Management, Providence University, Taichung 02912, Taiwan
* Correspondence: smping@dlmu.edu.cn

**Abstract:** Compared to multi-spectral imagery, hyperspectral imagery has very high spectral resolution with abundant spectral information. In underwater target detection, hyperspectral technology can be advantageous in the sense of a poor underwater imaging environment, complex background, or protective mechanism of aquatic organisms. Due to high data redundancy, slow imaging speed, and long processing of hyperspectral imagery, a direct use of hyperspectral images in detecting targets cannot meet the needs of rapid detection of underwater targets. To resolve this issue, a fast, hyperspectral underwater target detection approach using band selection (BS) is proposed. It first develops a constrained-target optimal index factor (OIF) band selection (CTOIFBS) to select a band subset with spectral wavelengths specifically responding to the targets of interest. Then, an underwater spectral imaging system integrated with the best-selected band subset is constructed for underwater target image acquisition. Finally, a constrained energy minimization (CEM) target detection algorithm is used to detect the desired underwater targets. Experimental results demonstrate that the band subset selected by CTOIFBS is more effective in detecting underwater targets compared to the other three existing BS methods, uniform band selection (UBS), minimum variance band priority (MinV-BP), and minimum variance band priority with OIF (MinV-BP-OIF). In addition, the results also show that the acquisition and detection speed of the designed underwater spectral acquisition system using CTOIFBS can be significantly improved over the original underwater hyperspectral image system without BS.

**Keywords:** constrained-target optimal index factor band selection (CTOIFBS); hyperspectral image; underwater spectral imaging system; underwater hyperspectral target detection; band selection (BS); constrained energy minimization (CEM)

## 1. Introduction

Underwater target detection using the images acquired by traditional red-green-blue (RGB) cameras has become more and more mature where traditional image processing methods [1,2] and target detection algorithms based on deep learning, such as Faster Region-based Convolutional Neural Networks (Faster R-CNN) [3] and You Only Look Once (YOLO) [4], have been widely applied to underwater target detection. In an ideal underwater imaging environment, the detection speed and

accuracy of various algorithms can reach a high level of performance. However, the traditional RGB image detection technology suffers from a series of problems. When the underwater imaging environment is poor and marine animals have their protective color mechanism, it is difficult to detect and identify targets of interest effectively from the complex background [5,6].

Hyperspectral imaging technology can provide a higher spectral resolution than RGB images, and its band coverage can range from ultraviolet, visible, near-infrared to mid-infrared bands and provides wealthy spectral information. Hyperspectral data is generally acquired by hundreds of contiguous narrow spectral bands, which can resolve the problems encountered in traditional RGB image detection technology and also make it have a good ability to identify targets and distinguishing similar targets. Classical hyperspectral target detection algorithms include an anomaly detector developed by Reed and Xiaoli, called the RXD algorithm [7], kernel RXD (KRXD) algorithm [8], orthogonal subspace projection (OSP) algorithm [9], and constrained energy minimization (CEM) algorithm [10]. Among them, CEM is a subpixel target detection algorithm that has been shown to be an effective and promising technique when only the target spectrum of interest is known and the background spectrum is unknown. So, it is quite suitable for target detection in a complicated underwater background and environment with insufficient prior knowledge.

At the present time, only a few studies on hyperspectral underwater target detection are available in the literature and most of them mainly focused on three aspects. First, in order to ensure that the hyperspectral imager can accurately extract key information in the complex marine environment when collecting underwater images, the designed key technologies are different. Second, since a hyperspectral image has a large number of spectral bands with very high spectral resolution, it has good target recognition ability. However, this is also traded for a slow imaging speed, enormous data volume, long transmission cycle, and slow calculation speed, all of which cannot be suitable for the remote operated vehicle (ROV) platform and real-time underwater target detection [11,12]. Third, hyperspectral underwater target detection technology tends to have strategic significance in both military and economic aspects. So, the degree of technological openness is extremely limited.

Because of the low imaging speed and long processing time of underwater hyperspectral images, the current research on underwater spectral imaging and detection is mainly focused on the detection of underwater pipelines, the distribution and species detection of underwater plants and microorganisms, etc. [13–17], but the capability of real-time detection is low. Some researchers have complied a spectral library for recognition and detection of different underwater targets. Kuniaki Uto et al. [18] classified the objects of interest by measuring their average spectral curves of cauliflower and sand to calculate their resultant correlation coefficient. Tegdan [19] et al. used a spectral library of some known objects of interest to achieve automatic recognition of other objects. An underwater hyperspectral imaging (UHI) system, jointly developed by Norwegian company Ecotone and Norwegian Underwater application robotics, is an optimized underwater hyperspectral imaging system, which can be used for underwater hyperspectral remote sensing. This system is capable of collecting information in the full color spectrum (370–800 nm).

In this paper, sea cucumbers are selected as our primary underwater targets due to its economic value on gross domestic product (GDP) growth in Dalian, China. Marine aquaculture is one of Dalian's pillar industries with an annual output value of more than 3.5 billion US dollars. Sea cucumbers are the major seafood products to account for the most revenue. At present, the main methods for fishing sea cucumbers, abalone, and other sea treasures rely on diver operation and submarine trawl operation. However, such diver operation is inefficient, and the deep-sea environment is extremely harmful to the health of divers. On the other hand, the submarine trawl operation generally causes severe damage to the underwater ecological environment. Therefore, autonomous fishing of seafood using underwater robots has become the most effective solution, and the rapid detection of underwater objects is a key issue that needs to be solved urgently.

Comparing to other targets, sea cucumber detection has more difficulty and greater challenges because sea cucumbers have a strong protective color mechanism. It is difficult to observe using

color and texture characteristics when ordinary RGB cameras are used for underwater observation. However, the sea cucumber exhibits relatively obvious reflectance characteristics in some special bands, which is the exact reason why we use hyperspectral technology to solve this problem.

The methods described above can effectively apply hyperspectral imaging technology to underwater biological classification and detection but cannot achieve real-time detection of underwater targets [20]. For the target to be detected, if its sensitive bands can be selected for detection in advance, the image processing speed can be increased to satisfy the real-time requirements. Gleason [21] found that the bands of 546, 568, and 589 nm could more easily separate corals and algae from other background objects. So, a multi-spectral camera could be constructed by six bands for fast acquisition of images for target detection. Experiments show that compared to the traditional RGB cameras, the six-band multi-spectral cameras had better performance in detecting submarine corals. However, the selected bands used for coral detection in the experiments were obtained as a by-product of other experiments, which are not applicable to other underwater targets and are not universal. Therefore, a reliable BS method needs to be designed so that it can select representative band subsets for different targets.

The researchers put forward some effective methods for BS. For example, information divergence (ID) selects bands according to the difference between the probability distributions of a measured band and its corresponding Gaussian probability distribution. The maximum-variance principal component analysis (MVPCA) developed in [22] first performed PCA transformation on the original data and then constructed the loading factor matrix from the obtained eigenvectors and eigenvalues. The priority of a band was determined by the variance of its corresponding loading factor. However, the bands selected according to such band prioritization methods were usually highly correlated. By factoring band correlation into consideration, the optimal index factor (OIF) [23] method was developed to find the largest OIF index. Yang et al. [24] proposed a BS method based on linear prediction, which used linear prediction as a similarity measure to find the next least similar band by sequential forward selection. All of the described methods select band subsets in accordance with the characteristics of the data itself and are not designed to select an optimal band subset for a specific target.

For target detection, Yuan et al. [25] proposed a multigraph determinantal point process (MDPP) model to effectively search for discriminative band sets. Wang [26] proposed the multi-band selection (MBS) method, which did not require prioritizing the bands but relied on a specific application to select desired bands. Based on the concept of CEM, Geng [27] proposed a sparse constrained band selection (SCBS), which is convenient for solving the global optimal solution and avoids the complicated subset search process. Wang et al. [28] proposed a new multi-target detection BS method, MinV-BP, which minimized the variance generated by the target of interest to measure the priority of the band.

This paper proposes a real-time detection method for hyperspectral underwater targets based on BS. First of all, in order to solve the problems suffering from a large amount of redundant data and slow acquisition and processing speed of hyperspectral image data, a BS method is designed in combination with MinV-BP [28] and OIF [23] to select an optimal band subset with strong ability in characterizing specific targets, called constrained-target OIF band selection (CTOIFBS). Then, an underwater multi-spectral sensor composed of the selected bands is particularly designed to collect images to overcome the difficulty of long transmission time of the complete hyperspectral image. Finally, CEM is used to detect underwater targets. The proposed CTOIFBS not only can extract a set of bands more suitable for specific targets to improve detection performance but can also meet the real-time requirements of underwater image acquisition.

## 2. Materials and Methods

### 2.1. MinV-BP

The idea of the Minimum Variance Band Prioritization (MinV-BP) is based on CEM, which was derived from the linearly constrained minimum variance beamformer in the field of digital signal

processing. It detects signals in a specific direction and minimizes signal interference in other directions, thereby achieving target detectability from the image and suppressing the background [10].

Suppose $\{\mathbf{r}_1, \mathbf{r}_2, \ldots, \mathbf{r}_N\}$ is a hyperspectral image with $N$ pixels. $N$ is the total number of pixels in the image. Each pixel, $\mathbf{r}_i = (\mathbf{r}_{i1}, \mathbf{r}_{i2}, \ldots, \mathbf{r}_{iL})^T$, is an $L$-dimensional column vector, where $L$ is the number of bands. Define $\mathbf{d}$ as the target spectral signal to be detected, which is known prior information. The purpose of CEM is to design a linear FIR filter $\mathbf{w} = [\mathbf{w}_1, \mathbf{w}_2, \ldots, \mathbf{w}_L]^T$ so that its output energy is minimized under the constraint term (1):

$$\mathbf{d}^T \mathbf{w} = \sum_{l=1}^{L} \mathbf{d}_l \mathbf{w}_l = 1 \tag{1}$$

where $\mathbf{w} = [\mathbf{w}_1, \mathbf{w}_2, \ldots, \mathbf{w}_L]^T$ is an $L$-dimensional column vector formed by the filter coefficient. Suppose the output of the FIR filter corresponding to the input pixel $\mathbf{r}_i$ is $\mathbf{y}_i$ defined in Equation (2):

$$\mathbf{y}_i = \sum_{i=1}^{L} \mathbf{w}_l \mathbf{r}_{il} = \mathbf{w}^T \mathbf{r}_i = \mathbf{r}_i^T \mathbf{w} \tag{2}$$

Then, for all input $\{\mathbf{r}_1, \mathbf{r}_2, \ldots, \mathbf{r}_N\}$, the average energy of the filter output is:

$$E = \frac{1}{N} \sum_{i=1}^{N} \mathbf{y}_i^2 = \frac{1}{N} (\mathbf{r}_i^T \mathbf{w})^T \mathbf{r}_i^T \mathbf{w} = \frac{1}{N} \sum_{i=1}^{N} \mathbf{w}^T \mathbf{r}_i \mathbf{r}_i^T \mathbf{w} = \mathbf{w}^T \left( \frac{1}{N} \sum_{i=1}^{N} \mathbf{r}_i \mathbf{r}_i^T \right) \mathbf{w} = \mathbf{w}^T \mathbf{R} \mathbf{w} \tag{3}$$

where $\mathbf{R} = \left( \frac{1}{N} \right) \sum_{i=1}^{N} \mathbf{r}_i \mathbf{r}_i^T$ represents the sample autocorrelation matrix of the $L \times L$ dimension. CEM can be expressed as the following linear constrained optimization problem:

$$\min_{\mathbf{w}} \{E\} = \min_{\mathbf{w}} \{\mathbf{w}^T \mathbf{R} \mathbf{w}\} \ s.t. \mathbf{d}^T \mathbf{w} = 1 \tag{4}$$

By using the Lagrange multiplier method, the optimal solution and CEM error of Equation (4) are obtained as follows:

$$\mathbf{w}_{CEM} = \frac{\mathbf{R}^{-1} \mathbf{d}}{\mathbf{d}^T \mathbf{R}^{-1} \mathbf{d}} \tag{5}$$

and:

$$\min_{\mathbf{w}} \mathbf{w} \mathbf{R}^{-1} \mathbf{w} = \left( \mathbf{w}^{CEM} \right)^T \mathbf{R}^{-1} \mathbf{w}^{CEM} = \left( \mathbf{d}^T \mathbf{R}^{-1} \mathbf{d} \right)^{-1} \tag{6}$$

The CEM filter is obtained from Equation (5):

$$\delta_{CEM}(\mathbf{r}) = (\mathbf{w}_{CEM})^T \mathbf{r} = \left( \frac{\mathbf{R}^{-1} \mathbf{d}}{\mathbf{d}^T \mathbf{R}^{-1} \mathbf{d}} \right)^T \mathbf{r} = \frac{\mathbf{d}^T \mathbf{R}^{-1} \mathbf{r}}{\mathbf{d}^T \mathbf{R}^{-1} \mathbf{d}} \tag{7}$$

The CEM operator is applied to every pixel in the image to minimize the output energy caused by other unknown signals so that the target $\mathbf{d}$ of interest can be detected to achieve the purpose of detection.

According to the CEM algorithm, single band minimum variance band prioritization (MinV-BP) can further use the variance generated by the target of interest to measure the priority of the band to obtain the band with the best characterization ability for the specific target. Suppose $\{\mathbf{b}_l\}_{l=1}^{L}$ is the band set of hyperspectral image, where $\mathbf{b}_l$ is a column vector, $\mathbf{b}_l = (b_{l1}, b_{l2}, \cdots, b_{lN})^T$, representing the image of the $l$-th band. $\{b_{li}\}_{i=1}^{N}$ is the set of all $N$ pixels on the $l$-th band image $\mathbf{b}_l$. According to the CEM error derived from Equation (6), MinV-BP is defined as:

$$V(\mathbf{b}_l) = \left( \mathbf{d}_{\mathbf{b}_l}^T \mathbf{R}_{\mathbf{b}_l}^{-1} \mathbf{d}_{\mathbf{b}_l} \right)^{-1} \tag{8}$$

Using Equation (8), MinV-BP can obtain the band priority sequence for the target of interest. Where, the smaller the variance, the higher the priority. The larger the variance, the lower the priority.

In short, the advantage of MinV-BP is that it can give higher priority to the band with strong target characterization ability through the minimum variance criterion. However, when MinV-BP prioritizes the bands, it only considers the ability of the bands to represent the target vector but does not consider the strong correlation and redundancy between the bands. As a result, the bands with high priority in the resulting sequence are largely adjacent bands with a strong correlation. Therefore, how to de-correlate the priority bands and obtain a band set with weak correlation and stronger discrimination ability is a subsequent problem to be solved.

### 2.2. OIF

Chavez et al. [23] proposed the optimum index factor (OIF) defined as:

$$\text{OIF} = \sum_{i=1}^{L} \mathbf{S}_i / \sum_{i=1}^{L} \sum_{j=i+1}^{L} \left| \mathbf{R}_{ij} \right| \tag{9}$$

to evaluate the amount of information in a dataset where $\mathbf{S}_i$ and $\mathbf{R}_{ij}$ represent the standard deviation of the *l*-th band and the correlation coefficient between band *i* and *j*, respectively, and *L* is the total number of bands. The standard deviation is used to represent the amount of image information. Based on the ratio of the amount of information in the band set to the correlation coefficient between the bands defined by:

$$\mathbf{R}_{ij} = \frac{\mathbf{S}_{ij}^2}{\mathbf{S}_i \times \mathbf{S}_j} \tag{10}$$

A band subset with a large amount of information and a small correlation can be selected as a band subset. In Equation (10), $\mathbf{S}_{ij}$ represents the covariance of bands *i* and *j*, and:

$$\mathbf{S}_{ij}^2 = Cov(i, j) = \frac{1}{n} \sum_{w=1}^{n} (\mathbf{x}_{iw} - \overline{\mathbf{x}}_i)(\mathbf{y}_{jw} - \overline{\mathbf{y}}_j) \tag{11}$$

where $\mathbf{x}_i$ represents the spectral grayscale value for the *i*-th band; $\mathbf{x}_{iw}$ represents the gray value of the *w*-th pixel in the *i*-th band; $\overline{\mathbf{y}}_i$ represents the spectral grayscale value for the *j*-th band; $\mathbf{y}_{jw}$ represents the gray value of the *w*-th pixel in the *j*-th band; *N* represents the number of pixels in a single band and *n* is the *n*-th pixel in the band, $1 \leq n \leq N$.

In other words, for a hyperspectral image containing *L* bands, the standard deviation of the single-band image and the correlation coefficient matrix of each band are calculated first, and then the OIF index corresponding to all possible band subsets are calculated subsequently, and the optimal band subset is finally selected according to the index value.

### 2.3. Constrained-Target OIF Band Selection

Hyperspectral data generally have very high band correlation and data redundancy. In order to mitigate this problem, a BS method with target constraints, called constrained-target optimum index factor BS (CTOIFBS), is developed in this paper. It first prioritizes all bands by MinV-BP to obtain a band priority sequence. The smaller the variance, the higher the priority of the band, and the stronger the ability of the band to represent the target. It is then followed by estimating virtual dimensionality (VD) [10,29–31] to determine the required number of bands, $n_{\text{BS}}$, where VD is defined as the number of spectrally distinct signal sources present in the data that can effectively characterize the hyperspectral data from a perspective view of target detection and classification. In this case, the first *n* bands with higher priorities in the sequence are clustered into $n_{\text{BS}}$ clusters by a K-means method to remove the band correlation. As a result, the band correlation in the same cluster will be high, while the band correlation between different clusters will be low. Finally, a band is selected from each cluster to form a

band subset. The OIF value of the band subset is then calculated. The band subset with the largest OIF value is selected as the best band subset. The CTOIFBS process is as follows.

---

**Algorithm CTOIFBS**

---

Input: Hyperspectral image data $\boldsymbol{\Omega}$

Output: The optimal band set $\boldsymbol{\Omega}^*_{n_{\mathrm{BS}}}$

1. According to (8), all bands $\mathbf{b}_l$ in $\boldsymbol{\Omega}$ are ranked to obtain the priority sequence of bands, $\mathbf{b}_{l_1} > \mathbf{b}_{l_2} > \cdots > \mathbf{b}_{l_L}$, where $\mathbf{b}_{l_j} > \mathbf{b}_{l_k} \Leftrightarrow \mathrm{V}\left(\mathbf{b}_{l_j}\right) < \mathrm{V}(\mathbf{b}_{l_k})$, the notation ">" is used to indicate "superior to".

2. The required number of bands $n_{\mathrm{BS}}$ is determined by VD.

3. The first $n$ bands of priority sequence obtained in step 1 were divided into $n_{\mathrm{BS}}$ bands set $\boldsymbol{\Omega}_k$ by K-means, where $\boldsymbol{\Omega}_k = \{\mathbf{b}_1^k, \mathbf{b}_2^k, \ldots, \mathbf{b}_{n_k}^k\}$, $1 \leq k \leq n_{\mathrm{BS}}$, $n_k$ denotes the number of bands included in $\boldsymbol{\Omega}_{n_{\mathrm{BS}}}$, $n = \sum_{i=1}^{n_{\mathrm{BS}}} n_k$.

4. Combining the bands in $\boldsymbol{\Omega}_k$, $\boldsymbol{\Omega}^* = \{\boldsymbol{\Omega}_1 \times \cdots \times \boldsymbol{\Omega}_K\}$, where "×" stands for cartesian product. $\boldsymbol{\Omega}^*$ contains $M$ band sets, $M = n_1 \times n_2 \times \cdots n_K$. Then calculate the OIF value of each band set in $\boldsymbol{\Omega}^*$.

5. The maximum OIF value is selected as the optimal band set $\boldsymbol{\Omega}^*_{n_{\mathrm{BS}}}$.

---

A flowchart implementing CTOIFBS is shown in Figure 1.

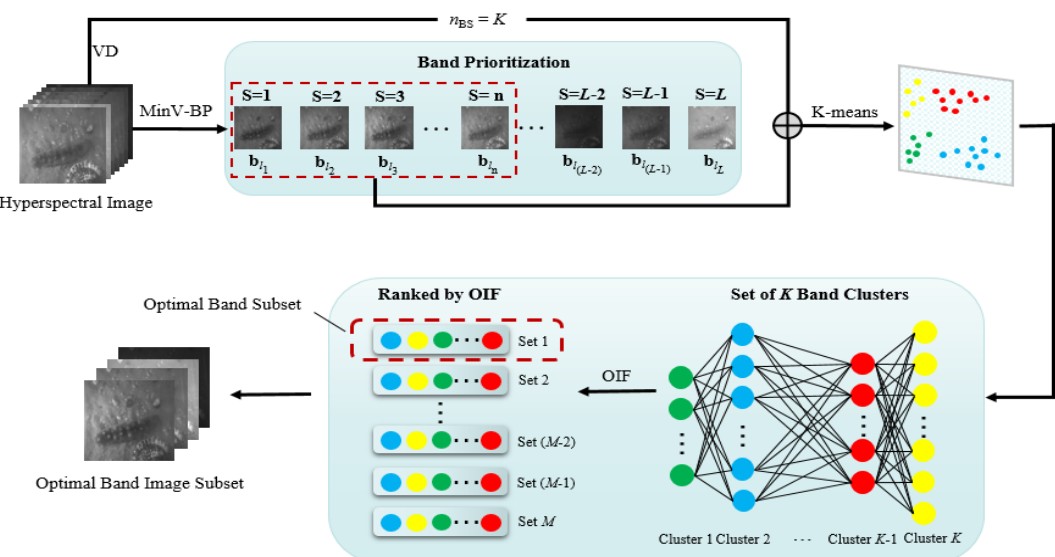

**Figure 1.** A flowchart of implementing CTOIFBS (constrained-target optimal index factor band selection).

Using the MinV-BP criterion, a band priority sequence for the target of interest can be obtained, and then bands with strong characterization of the target can be selected from all the band sequence. However, there is still a problem, which is high inter-band correlation in this band sequence. OIF takes two factors into account: variance and correlation coefficient. Theoretically, the optimal band subset with large information amount and small inter-band correlation can be obtained by optimizing the priority sequence of the band using OIF. However, it has been found in experiments that the use of OIF alone to process band priority sequences was not effective since a band subset with high

correlation will still be selected. This is because OIF strives to make the standard deviation of the selected bands as large as possible, while the correlation coefficient between the bands is as small as possible. Unfortunately, it is difficult to achieve the best of both measures [15]. Therefore, instead of selecting the first *n* bands of the priority sequence directly by the OIF index as a band subset, CTOIFBS is developed to use clusters to perform band de-correlation prior to using OIF. That is, the selected candidate bands are divided into several subsets to further reduce the band correlation and band redundancy. The advantages of such cluster-based band de-correlation have two advantages. One is the pre-grouping process, which reduces the total number of band subset to be compared so that computational complexity can be greatly reduced. The other is clustering by a K-means method in advance to effectively remove band redundancy so as to improve subsequent detection performance.

## *2.4. Underwater Spectral Imaging System*

Using an underwater spectrum camera composed of a best-selected band subset to collect the target image can greatly reduce data redundancy and solve the problem of long transmission time of a complete hyperspectral image. However, due to the complicated underwater imaging environment on the one hand and the difficulty in finding the proper loader or vehicle on the other hand, the development of underwater spectral imaging technology is still far from that of atmospheric spectral imaging. Therefore, how to design a suitable underwater spectral imaging (USI) system is the very key to success in realizing the rapid detection of hyperspectral underwater targets.

The core of the spectral imaging system is the optical splitting system. The spectroscopic techniques currently being used are based on dispersion, filtering, and interferometry, and commonly used optical splitting components include gratings, prisms, and various filters. This paper develops a filter wheel spectral camera to collect spectral images. There are several reasons. First of all, it has a wheel with multiple single band-pass filters to collect spectral information of different bands, which is suitable for the case of fewer bands needed. Second, a narrow band filter has a high transmittance, so it is suitable for the special light conditions under water. Third, it adapts to different filter combinations that can be changed according to different objects. Fourth, this type of camera is much cheaper than the commonly used liquid crystal tunable filter (LCTF) spectral camera.

Therefore, this paper builds an underwater spectral imaging system based on a filter wheel spectral camera, as shown in Figure 2. Its main components include a FLIR Blackfly S USB3 CCD camera and its corresponding lens, electric filter wheel, and single band-pass filters with the wavelengths between 400 and 830 nm at intervals of 10 nm. These filters have a bandwidth of 14nm and a cut-off depth of OD3 and a single chip microcomputer for controlling the camera and filter wheel. All the above parts are packed in a watertight enclosure. This system uses electric filter wheels to collect single-band images in different bands and synthesize the target's spectral image. It is also possible to obtain spectral images of different band subsets by replacing the filter combinations on the filter wheel. It is important to note that the spectral filter wheel designed is not limited to the USI system and can be applicable to various beam splitters, such as LCTF, acousto-optic tunable filter (AOTF), or spectral filter array (SFA) according to their application scenarios and costs.

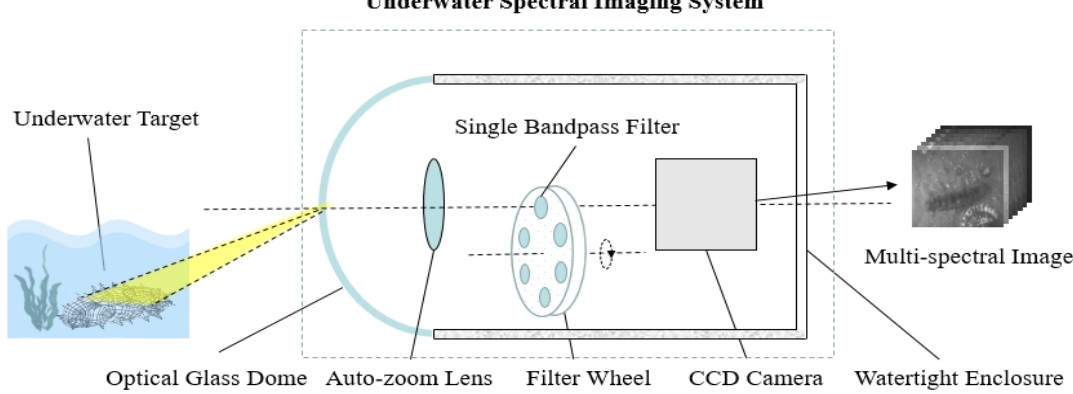

**Figure 2.** Diagram of the underwater spectral imaging system.

## 3. Results and Discussion

The experiments conducted in this section are divided into three parts. The first part is to validate the performance of the CTOIFBS on a real hyperspectral image, i.e., hyperspectral digital imagery collection experiment (HYDICE) data. A second part is to apply CTOIFBS to real underwater hyperspectral data and to use the calibrated image to select a band subset to validate the CTOIFBS used for the test image. A third part is to design an underwater spectral imaging system to be used to collect the band images of underwater targets according to bands selected by CTOIFBS for detection to verify the feasibility of the USI system for rapid detection of underwater targets and the superiority of CTOIFBS to other BS methods. To further justify the three BS methods, UBS, MinV-BP, and MinV-BP-OIF along with full bands are compared in the experiments where MinV-BP-OIF uses OIF to directly select the optimal band subset for the first $n$ bands selected by MinV-BP. The main difference between CTOIFBS and MinV-BP-OIF is that prior to calculating the OIF value, CTOIFBS uses the K-means method to divide the first $n$ bands selected by MinV-BP into $n_{BS}$ spectral low-relevance clusters. Then, CTOIFBS combines each band from various clusters to form a band subset and then selects a band subset with the largest OIF value as the desired band subset. Comparing to MinV-BP-OIF, the correlation among the bands selected by CTOIFBS is lower than MinV-BP-OIF. In addition, the required number of bands for HYDICE and real underwater hyperspectral data of sea cucumbers were determined by virtual dimensionality (VD) [10,29], which are six and five, respectively. Finally, visual inspection and quantitative analysis are also used to analyze and compare the performance of various BS methods.

Specifically, a 3D receiver operating characteristic (ROC) analysis-based quantitative analysis developed in [32,33] was conducted by calculating the area under the curve (AUC) for the 2D ROC curves of $(P_D, P_F)$, $(P_D, \tau)$, and $(P_F, \tau)$ widely used in target detection where $P_D$ and $P_F$ represent the detection probability and the false alarm probability defined in [34], respectively, which were produced by using a different $\tau$ range from 0 to 1 to binarize the normalized detection result. The AUC values of $(P_D, P_F)$, $(P_D, \tau)$, and $(P_F, \tau)$ were used to measure the overall detection performance, target detection capability, and background suppression ability of a detector, respectively. It should be noted that the higher the AUC values of $(P_D, P_F)$ and $(P_D, \tau)$ are, the better the detection performance of the detector is. Conversely, the smaller the AUC value of $(P_F, \tau)$, the better the suppression ability of the background.

### 3.1. Real HYDICE Image

This real HYDICE scene has been widely used in target detection. It has a spatial resolution of 1.56 m and contains 169 spectral bands with a size of 64 × 64. There are 15 panels divided into five types of targets, $\mathbf{p}_1$, $\mathbf{p}_2$, $\mathbf{p}_3$, $\mathbf{p}_4$, and $\mathbf{p}_5$, which are distributed on each row with three different sizes, 3 × 3 m, 2 × 2 m, and 1 × 1 m, respectively shown in Figure 3a. Figure 3b shows their precise spatial

locations with the pixels in yellow (Y pixels), indicating panel pixels mixed with the BKG. In addition, there are a total of 19 panel pixels highlighted by red, which are the target pixels to focus on.

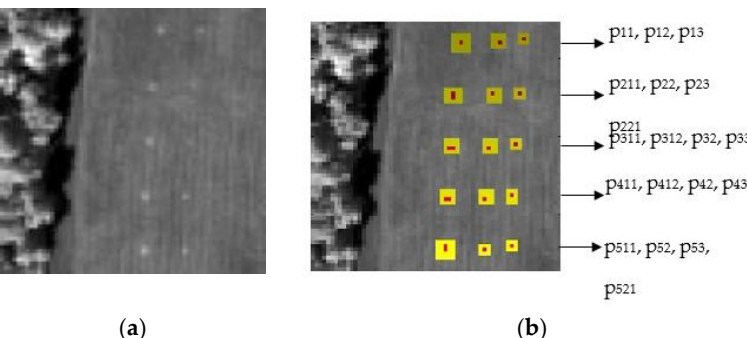

<div align="center">(<b>a</b>)　　　　　　　　　　　　　　　　　　　(<b>b</b>)</div>

**Figure 3.** (**a**) Hyperspectral digital imagery collection experiment (HYDICE) scene. (**b**) Ground truth map of the 15 panels.

Table 1 shows the band subsets selected by four BS methods along with full bands for target $p_1$, $p_2$, $p_3$, $p_4$, and $p_5$ in the HYDICE image. Unlike UBS, which is independent of targets, when the desired targets are different, the bands selected by three BS methods for target detection, MinV-BP, MinV-BP-OIF, and CTOIFBS, are also different. Figure 4 shows the detection results of each target under different sets of bands using CEM. From the intuitive detection results, it can be seen that the detection results are best when using the full bands with the background well suppressed. When using the set of bands selected by MinV-BP and UBS to detect targets, undesired targets respond strongly and are clearly detected. Moreover, the detection results of UBS showed that the band selected by UBS had a weak suppression ability on the background. Finally, compared with the MinV-BP-OIF and CTOIFBS methods, it can be obtained that CTOIFBS has a better ability to detect targets and has a good background suppression effect.

**Table 1.** Optimal band subsets selected by four BS (band selection) methods along with full bands.

| Target | Method | Band Set |
|---|---|---|
| | Full bands | 1:1:169 |
| | UBS | 1 29 57 86 114 142 |
| $p_1$ | MinV-BP | 169 122 123 168 167 166 |
| | MinV-BP-OIF | 133 134 98 99 135 100 |
| | CTOIFBS | 122 169 131 98 162 149 |
| $p_2$ | MinV-BP | 122 169 123 132 133 131 |
| | MinV-BP-OIF | 136 98 137 138 99 100 |
| | CTOIFBS | 159 100 137 128 122 98 |
| $p_3$ | MinV-BP | 122 123 132 133 124 131 |
| | MinV-BP-OIF | 52 53 51 54 99 100 |
| | CTOIFBS | 124 98 128 53 101 169 |
| $p_4$ | MinV-BP | 123 122 124 125 127 128 |
| | MinV-BP-OIF | 99 100 101 102 103 104 |
| | CTOIFBS | 99 103 124 137 127 122 |
| $p_5$ | MinV-BP | 122 123 124 125 126 168 |
| | MinV-BP-OIF | 127 134 135 98 138 145 |
| | CTOIFBS | 167 159 98 157 128 163 |

UBS: uniform band selection; MinV-BP: minimum variance band priority; MinV-BP-OIF: minimum variance band priority with OIF; CTOIFBS: constrained-target optimal index factor (OIF) band selection.

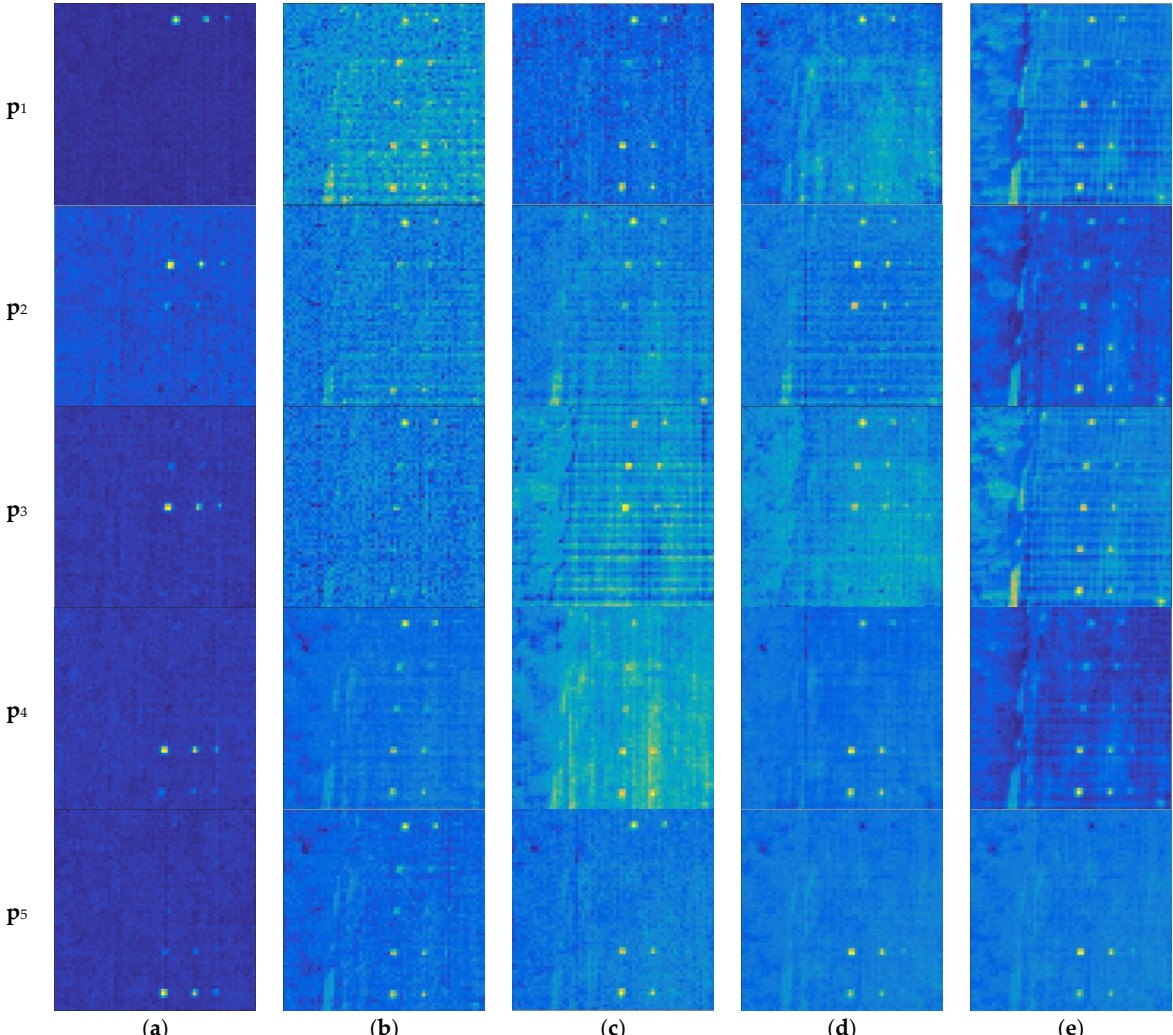

**Figure 4.** CEM (constrained energy minimization) detection map results using different band subsets selected by four BS (band selection) methods along with full bands: (**a**) Full bands; (**b**) MinV-BP: minimum variance band priority; (**c**) MinV-BP-OIF: minimum variance band priority with OIF; (**d**) CTOIFBS: constrained-target optimal index factor (OIF) band selection; (**e**) UBS: uniform band selection.

In addition to analyzing the performance of various BS methods by visual inspection, the experiment also performed quantitative analysis. Table 2 tabulates the AUC values of the five BS methods where the best and worst results are highlighted by red and green, respectively. The higher the AUC value, the better the detection, that is, the better the selected band subset to represent the target. As expected, the results using full bands were the best. However, among all the four BS methods, CTOIFBS generally outperformed the other three BS methods in terms of $(P_D, P_F)$. In order to further demonstrate the effectiveness of CTOIFBS, Table 3 ranks the AUC value of $(P_D, P_F)$ of various methods. The last row of Table 3 ranks the total target detection capability by the BS methods. The smaller the value, the better detection capability of the selected band subset. Among them, the value of full bands is five, ranking first, and the detection capability is the best. CTOIFBS scores 13, which is only worse than full bands. Although CTOIFBS is slightly inferior to using the full bands in detection performance, its transmission time and processing time are much lower than using the full bands due to the reduced data dimensionality. In addition, CTOIFBS performed better than MinV-BP, MinV-BP-OIF, and UBS assuming that the same number of selected bands was used.

**Table 2.** AUC (area under the curve) values of detection results for target $p_1$, $p_2$, $p_3$, $p_4$, and $p_5$ using four BS (band selection) methods along with full bands: (**a**) AUC values of detection results for target $p_1$; (**b**) AUC values of detection results for target $p_2$; (**c**) AUC values of detection results for target $p_3$; (**d**) The AUC values of detection results for target $p_4$; (**e**) AUC values of detection results for target $p_5$.

| | (a) | | |
|---|---|---|---|
| **Method** | **($P_D$, $P_F$)** | **($P_D$, $\tau$)** | **($P_F$, $\tau$)** |
| Full bands | 0.9993 | 0.6017 | 0.0328 |
| MinV-BP | 0.8179 | 0.7017 | 0.3120 |
| MinV-BP-OIF | 0.9794 | 0.5817 | 0.1603 |
| CTOIFBS | 0.9274 | 0.6583 | 0.2418 |
| UBS | 0.9784 | 0.5550 | 0.2170 |
| | (b) | | |
| **Method** | **($P_D$, $P_F$)** | **($P_D$, $\tau$)** | **($P_F$, $\tau$)** |
| Full bands | 0.9998 | 0.8375 | 0.1012 |
| MinV-BP | 0.9847 | 0.4975 | 0.2378 |
| MinV-BP-OIF | 0.9943 | 0.5875 | 0.2534 |
| CTOIFBS | 0.9978 | 0.8200 | 0.2266 |
| UBS | 0.9837 | 0.3725 | 0.1160 |
| | (c) | | |
| **Method** | **($P_D$, $P_F$)** | **($P_D$, $\tau$)** | **($P_F$, $\tau$)** |
| Full bands | 0.9997 | 0.7425 | 0.0519 |
| MinV-BP | 0.9914 | 0.4850 | 0.2113 |
| MinV-BP-OIF | 0.9937 | 0.7650 | 0.3161 |
| CTOIFBS | 0.9968 | 0.5775 | 0.2989 |
| UBS | 0.9895 | 0.6775 | 0.2412 |
| | (d) | | |
| **Method** | **($P_D$, $P_F$)** | **($P_D$, $\tau$)** | **($P_F$, $\tau$)** |
| Full bands | 0.9998 | 0.7750 | 0.0546 |
| MinV-BP | 0.9953 | 0.5700 | 0.1868 |
| MinV-BP-OIF | 0.9944 | 0.8150 | 0.3585 |
| CTOIFBS | 0.9985 | 0.7775 | 0.1918 |
| UBS | 0.9954 | 0.5925 | 0.1007 |
| | (e) | | |
| **Method** | **($P_D$, $P_F$)** | **($P_D$, $\tau$)** | **($P_F$, $\tau$)** |
| Full bands | 0.9998 | 0.7000 | 0.0495 |
| MinV-BP | 0.9960 | 0.6175 | 0.1574 |
| MinV-BP-OIF | 0.9952 | 0.7500 | 0.2148 |
| CTOIFBS | 0.9935 | 0.5625 | 0.2187 |
| UBS | 0.9954 | 0.7000 | 0.0988 |

MinV-BP: minimum variance band priority; MinV-BP-OIF: minimum variance band priority with OIF; CTOIFBS: constrained-target optimal index factor (OIF) band selection; UBS: uniform band selection.

**Table 3.** Order of the AUC (area under the curve) values of ($P_D$, $P_F$) of four BS (band selection) methods along with full bands.

| | **Full Bands** | **MinV-BP** | **MinV-BP-OIF** | **CTOIFBS** | **UBS** |
|---|---|---|---|---|---|
| $p_1$ | 1 | 5 | 2 | 4 | 3 |
| $p_2$ | 1 | 4 | 3 | 2 | 5 |
| $p_3$ | 1 | 4 | 3 | 2 | 5 |
| $p_4$ | 1 | 4 | 5 | 2 | 3 |
| $p_5$ | 1 | 2 | 4 | 5 | 3 |
| SUM | 5 | 19 | 17 | 13 | 19 |

MinV-BP: minimum variance band priority; MinV-BP-OIF: minimum variance band priority with OIF; CTOIFBS: constrained-target optimal index factor (OIF) band selection; UBS: uniform band selection.

### 3.2. Underwater Hyperspectral Image

In this section, real hyperspectral data were collected and conducted for sea cucumber detection to validate the performance of CTOIFBS. To demonstrate the effectiveness of CTOIFBS, several state-of-the-art BS methods, full bands, UBS, MinV-BP, and MinV-BP-OIF are compared by experiments where the required number of bands is five determined by VD. Finally, detection results and quantitative analysis were used to analyze and compare the performance of various BS methods. Specifically, quantitative analysis was conducted by the area under the curve (AUC) widely used in target detection.

The data used in our experiments were underwater sea cucumber images collected by a hyperspectral imager, covering 256 bands with a spectral range of 0.4 to 1.05 nm. Due to the fast attenuation of infrared bands in underwater, the sensor could not collect enough information from infrared bands. So, part of the infrared bands (171–256) were removed, and only 1–170 bands were analyzed for experiments with a spectral coverage of 0.4~0.825 nm. Shown in Figure 5a,b are the RGB images of the calibrated data and their corresponding mask image, respectively.

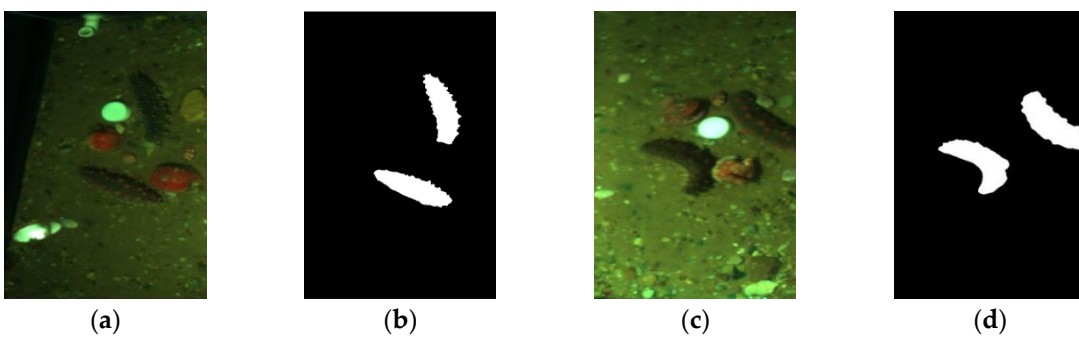

| (a) | (b) | (c) | (d) |

**Figure 5.** Sea cucumber data for experiments: (**a**) RGB image of calibrated data; (**b**) ground truth map of calibrated data; (**c**) RGB image of validated data; (**d**) ground truth map of validated data.

We have plotted the spectra for five types of ground features, including the sea cucumber, sand, pebble, clam, and scallop from calibrated data, as shown in Figure 5a, where the sea cucumber was selected as the target of interest and the other four features as the background. The obtained spectra were used to mark the spectral bands location (points) selected by the four BS methods in Table 4, which is shown in Figure 6 using red vertical dashed lines for visual inspection and comparison among correlation of the selected band sets.

**Table 4.** Band subsets selected by four BS (band selection) methods along with full bands.

| Method | Band Set |
| --- | --- |
| Full bands | 1:1:170 |
| MinV-BP | 170 168 43 46 36 |
| MinV-BP-OIF | 170 168 169 167 29 |
| CIOIFBS | 34 43 29 58 170 |
| UBS | 1 35 69 103 137 |

MinV-BP: minimum variance band priority; MinV-BP-OIF: minimum variance band priority with OIF; CTOIFBS: constrained-target optimal index factor (OIF) band selection; UBS: uniform band selection.

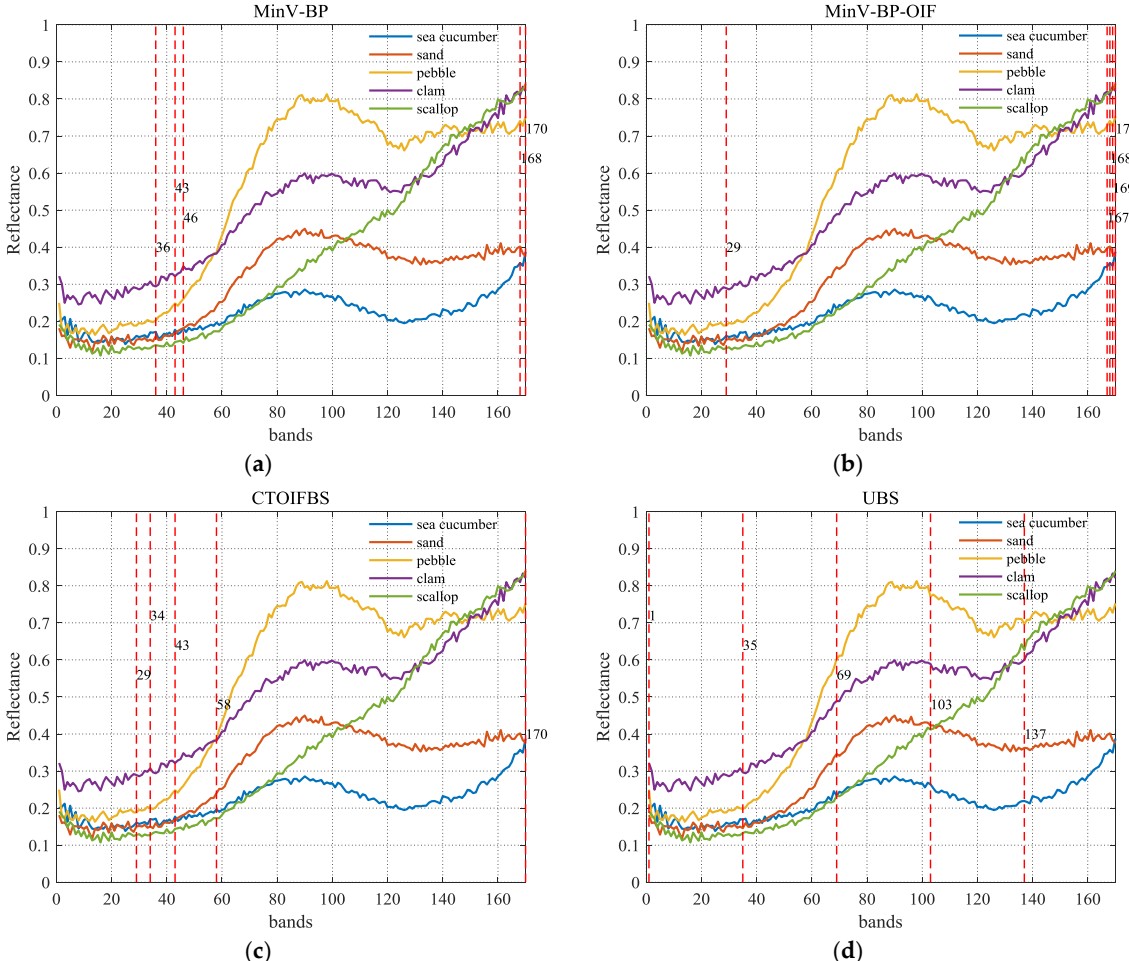

**Figure 6.** Bands selected by four BS (band selection) methods: (**a**) MinV-BP: minimum variance band priority; (**b**) MinV-BP-OIF: minimum variance band priority with OIF; (**c**) CTOIFBS: constrained-target optimal index factor (OIF) band selection; (**d**) UBS: uniform band selection.

On the one hand, comparing to MinV-BP and MinV-BP-OIF, CTOIFBS took the correlation among bands into consideration. As a result, the bands selected by CTOIFBS were more dispersed and contained more spectral information. On the other hand, although the distribution of band selected by UBS was more dispersed than the other three methods, the detection results were not satisfactory. This is because UBS did not consider the special relationship between the target and its selected bands. Consequently, it was unable to select bands pertaining to target information compared to the band set selected by CTOIFBS, which can effectively avoid high correlation between bands and can be further used to characterize targets of interest.

Table 5 shows the correlation coefficient among bands in each band subset selected by a different BS method where the greater the value between two bands in a band subset, the higher the correlation between these two bands. So, a better band subset should have less correlation among its bands. Furthermore, Table 6 shows the mean correlation coefficients among bands selected by different BS methods.

**Table 5.** Correlation coefficient matrices of the band subset selected by four BS (band selection) methods: (**a**) correlation coefficient matrix of the band subset selected by MinV-BP; (**b**) correlation coefficient matrix of the band subset selected by MinV-BP-OIF; (**c**) correlation coefficient matrix of the band subset selected by CTOIFBS; (**d**) correlation coefficient matrix of the band subset selected by UBS.

| (a) | | | | | |
|---|---|---|---|---|---|
| **Band no.** | **170** | **168** | **43** | **46** | **36** |
| 170 | 1 | | | | |
| 168 | 0.9913 | 1 | | | |
| 43 | 0.8840 | 0.8887 | 1 | | |
| 46 | 0.8958 | 0.9018 | 0.9928 | 1 | |
| 36 | 0.8420 | 0.8440 | 0.9809 | 0.9651 | 1 |
| (b) | | | | | |
| **Band no.** | **170** | **168** | **169** | **167** | **29** |
| 170 | 1 | | | | |
| 168 | 0.9913 | 1 | | | |
| 169 | 0.9918 | 0.9923 | 1 | | |
| 167 | 0.9904 | 0.9928 | 0.9919 | 1 | |
| 29 | 0.8016 | 0.8023 | 0.8021 | 0.8024 | 1 |
| (c) | | | | | |
| **Band no.** | **34** | **43** | **29** | **58** | **170** |
| 34 | 1 | | | | |
| 43 | 0.9711 | 1 | | | |
| 29 | 0.9920 | 0.9518 | 1 | | |
| 58 | 0.8261 | 0.9237 | 0.7890 | 1 | |
| 170 | 0.8269 | 0.8840 | 0.8016 | 0.8958 | 1 |
| (d) | | | | | |
| **Band no.** | **1** | **35** | **69** | **103** | **137** |
| 1 | 1 | | | | |
| 35 | 0.9212 | 1 | | | |
| 69 | 0.5197 | 0.7005 | 1 | | |
| 103 | 0.4206 | 0.6024 | 0.9694 | 1 | |
| 137 | 0.6408 | 0.7946 | 0.9405 | 0.9227 | 1 |

**Table 6.** Mean correlation coefficients of four BS (band selection) methods.

| Method | MinV-BP | MinV-BP-OIF | CTOIFBS | UBS |
|---|---|---|---|---|
| Mean correlation coefficient | 0.9186 | 0.9158 | 0.8862 | 0.7432 |

MinV-BP: minimum variance band priority; MinV-BP-OIF: minimum variance band priority with OIF; CTOIFBS: constrained-target optimal index factor (OIF) band selection; UBS: uniform band selection.

From Table 6, it can be seen that compared to the other two target-constrained BS methods, the mean correlation coefficient among the bands selected by CTOIFBS is the smallest, which validates the advantage of CTOIFBS in reducing correlation between bands during the BS. It is worth noting that although the mean correlation coefficient among the bands selected by UBS is the smallest, its detection results were poor due to its inability to select effective bands to characterize the target.

According to the band subsets selected by different BS methods in Table 4, their corresponding band images of the calibrated data shown in Figure 5a were synthesized. CEM was then used to detect sea cucumbers, and the detection results of using full bands and band subsets selected by four BS methods were shown in Figure 7. The brighter a pixel in the image is, the higher the probability that the pixel is considered to be more likely a target by the detector. It is also observed that the target pixels detected with a band set selected by UBS were not obvious and have been buried in the background.

Furthermore, the AUC values calculated in Table 7 were also used to quantitatively analyze the effect of different BS methods on detection performance where the best and worst results are highlighted by red and green, respectively. Comparing to the AUC values of ($P_D$, $P_F$), the full band was the best followed by CTOIFBS, MinV-BP-OIF, and MinV-BP, and finally, UBS.

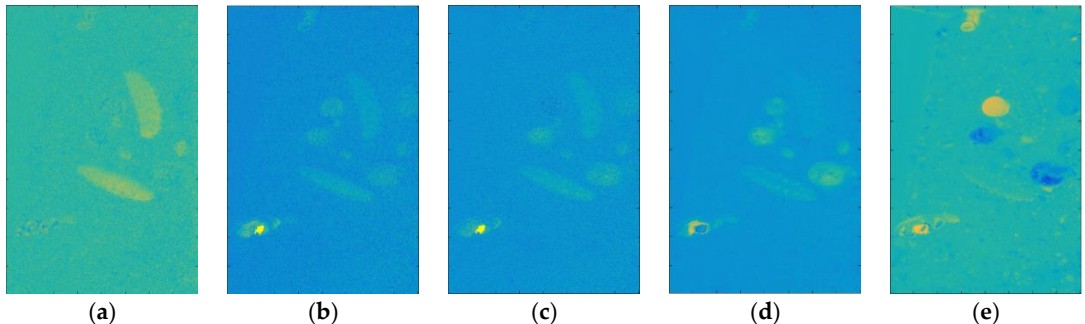

| (a) | (b) | (c) | (d) | (e) |

**Figure 7.** Detection results of the calibrated data of the RGB image and ground truth map shown in Figure 5a,b by full bands and four BS methods: (**a**) Full bands; (**b**) MinV-BP: minimum variance band priority; (**c**) MinV-BP-OIF: minimum variance band priority with OIF; (**d**) CTOIFBS: constrained-target optimal index factor (OIF) band selection; (**e**) UBS: uniform band selection.

**Table 7.** AUC (area under the curve) values of five BS (band selection) methods.

| Method | ($P_D$, $P_F$) | ($P_D$, $\tau$) | ($P_F$, $\tau$) |
|---|---|---|---|
| Full bands | 0.9022 | 0.5783 | 0.4917 |
| MinV-BP | 0.7315 | 0.3511 | 0.2998 |
| MinV-BP-OIF | 0.7577 | 0.3666 | 0.3145 |
| CTOIFBS | 0.7961 | 0.3522 | 0.3176 |
| UBS | 0.6148 | 0.4717 | 0.4601 |

MinV-BP: minimum variance band priority; MinV-BP-OIF: minimum variance band priority with OIF; CTOIFBS: constrained-target optimal index factor (OIF) band selection; UBS: uniform band selection.

In order to further validate the effectiveness of CTOIFBS in detecting underwater targets, an additional experimental image was also selected for testing the performance of various BS methods. Figure 8 shows the detection results of sea cucumbers on the test image using a set of bands selected in Table 4. Table 8 tabulates their AUC values where the best and worst results are highlighted by red and green, respectively. According to the AUC values of ($P_D$, $P_F$) in Table 8, the detection result of CTOIFBS was higher than that of other BS methods, MinV-BP, MinV-BP-OIF, and UBS using the same number of bands. As expected, the CTOIFBS result was only worse than that of using full bands. This proves that it is feasible to use the band subset selected by CTOIFBS for underwater target detection.

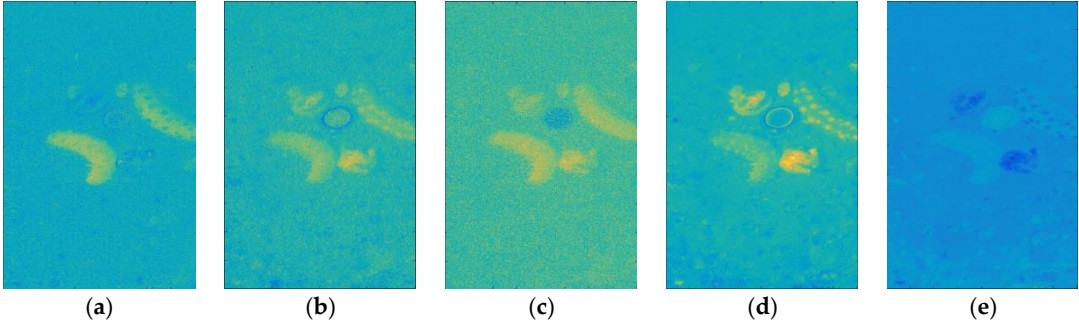

| (a) | (b) | (c) | (d) | (e) |

**Figure 8.** Detection results of the validated data of the RGB image and ground truth map shown in Figure 5c,d by full bands and four BS methods: (**a**) Full bands; (**b**) MinV-BP: minimum variance band priority; (**c**) MinV-BP-OIF: minimum variance band priority with OIF; (**d**) CTOIFBS: constrained-target optimal index factor (OIF) band selection; (**e**) UBS: uniform band selection.

**Table 8.** AUC (area under the curve) values of four BS (band selection) methods along with full bands.

| Method | $(P_D, P_F)$ | $(P_D, \tau)$ | $(P_F, \tau)$ |
|---|---|---|---|
| Full bands | 0.9396 | 0.5494 | 0.4036 |
| MinV-BP | 0.8482 | 0.5566 | 0.4542 |
| MinV-BP-OIF | 0.8442 | 0.6199 | 0.5181 |
| CTOIFBS | 0.9318 | 0.5349 | 0.4357 |
| UBS | 0.7460 | 0.3099 | 0.2914 |

MinV-BP: minimum variance band priority; MinV-BP-OIF: minimum variance band priority with OIF; CTOIFBS: constrained-target optimal index factor (OIF) band selection; UBS: uniform band selection.

The above real image sea cucumber image experiments also proved that it was feasible to use the band subset selected by CTOIFBS for underwater target detection. Although the detection result of CTOIFBS is slightly worse than that of using full bands, the acquisition and transmission speeds are considerably faster than using full bands because a smaller number of bands were used, and the smaller amount of image data is being processed. Table 9 shows the detection speeds of using full bands and CTOIFBS under the same experimental environment.

**Table 9.** Comparison of the average speed of two methods for detecting a single image.

| Method | Detection Speed (ms) |
|---|---|
| Full bands | 494 |
| CTOIFBS | 98 |

CTOIFBS: constrained-target optimal index factor (OIF) band selection.

From Table 9, the process of using full bands consumed a great deal of time, which was reflected in imaging, transmission, and processing. Under the effect of water flow, target movement, and other factors, a USI system needs to detect the target quickly. Obviously, a USI system using full bands cannot meet the requirement for rapid detection of an underwater target. In addition, studies have found that using full bands may incur an issue of the Hughes phenomenon [35], that is, high dimensionality may decrease the detection accuracy. Furthermore, the experiments further demonstrated that the detection results of CTOIFBS could be very close to that obtained using the full bands. With all things considered above, a USI system with full bands is not suitable for underwater rapid target detection.

*3.3. Underwater Spectral Imaging System*

In order to verify that the collected target spectral data by the constructed underwater spectral imaging (USI) system can accurately detect underwater targets, two experiments were set up in this section. The first experiment was conducted by comparing the hyperspectral data using the selected band subset to the multi-spectral data collected by the USI system using the same band subset under similar scenes to prove that the multi-spectral data collected by the USI system has consistent feature expression capability with the hyperspectral images. A second experiment was also conducted under the same scenes to compare the detection performance of data collected by the USI system using different BS methods to verify the detection capability of CTOIFBS.

3.3.1. First Experiment: Compatibility of USI to HSI

In order to show that the multi-spectral data collected by the USI system have the same feature expression ability as the hyperspectral images, the experiment collected the hyperspectral data and the filter bands corresponding to the band subset selected by CTOIFBS in similar scenes. Because the bands selected by CTOIFBS are 470, 480, 500, 540, and 830 nm, the band images corresponding to the hyperspectral data were extracted to form a band subset for subsequent target detection. Figure 9 shows the images collected by two methods and their corresponding detection results of sea cucumbers in similar scenes.

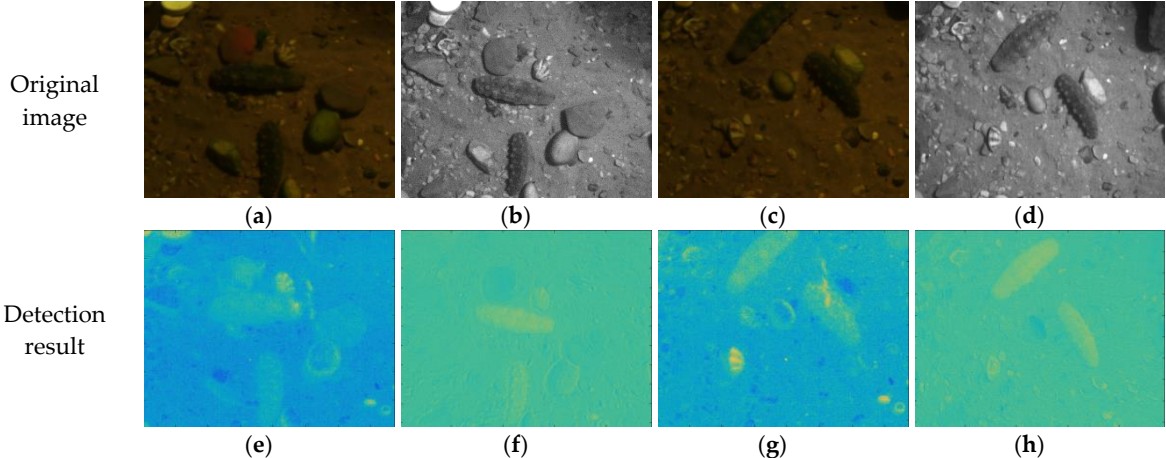

**Figure 9.** Images collected by two methods and corresponding detection map results in similar scenes. HSI-01 (**a**), HSI-02 (**c**) are hyperspectral images, USI-01 (**b**), USI-02 (**d**) are images collected by the USI system; (**e**), (**f**), (**g**), (**h**) are the detection results corresponding to (**a**), (**b**), (**c**), (**d**).

According to the detection results, both methods are capable of detecting sea cucumbers. From the performance of suppressing non-target pixels, although the image extracted from the HSI data can suppress the main background, which is sand, it has a high response to interference targets, such as stones and clams. By contrast, the data collected by the USI system can suppress non-target pixels more effectively. From the AUC values of ($P_D$, $P_F$) in Table 10, the AUC value detected using the data collected by the USI system is higher than that using HSI data, indicating that its ability to detect targets is higher. Of course, due to the difference in the performance of the sensors used by the two methods, this experiment may not have sufficient evidence to conclude that the detection results based on the data collected by the USI system must be better than the data using the corresponding band of HSI. Nevertheless, it can prove that the data collected using the USI system has the same feature expression ability as the hyperspectral images and can be used for underwater spectral data collection and target detection.

**Table 10.** AUC (area under the curve) values for CEM (constrained energy minimization) detection map results using four images shown in Figure 9.

| Data | ($P_D$, $P_F$) | ($P_D$, $\tau$) | ($P_F$, $\tau$) |
|---|---|---|---|
| HSI-01 | 0.7391 | 0.1459 | 0.0831 |
| USI-01 | 0.8673 | 0.0901 | 0.0318 |
| HSI-02 | 0.8451 | 0.2173 | 0.0940 |
| USI-02 | 0.9274 | 0.1329 | 0.0432 |

### 3.3.2. Second Experiment: USI System using CTOIFBS

This section uses the data collected by the USI system to compare the performance of the CTOIFBS with four BS methods. MinV-BP, MinV-BP-OIF, and UBS with their corresponding band subsets tabulated in Table 11. Then, the single-band images are collected by the USI system, as shown in Figure 10. Finally, the collected single-band images are integrated into multi-spectral image cubes for target detection. It should be noted that the single band image-constructed multi-spectral image data has indeed a spectral resolution of approximate 10 nm, and thus, the filters actually used are rounded to 10nm.

**Table 11.** Band subsets selected by four BS (band selection) methods.

| Methods | Selected Bands (nm) |
|---|---|
| MinV-BP | 825.6 820.4 506.0 513.5 489.1 |
| MinV-BP-OIF | 825.6 820.4 823.0 817.8 472.1 |
| CTOIFBS | 484.2 506.0 472.1 542.9 825.6 |
| UBS | 400.0 486.7 570.0 654.6 740.7 |

MinV-BP: minimum variance band priority; MinV-BP-OIF: minimum variance band priority with OIF; CTOIFBS: constrained-target optimal index factor (OIF) band selection; UBS: uniform band selection.

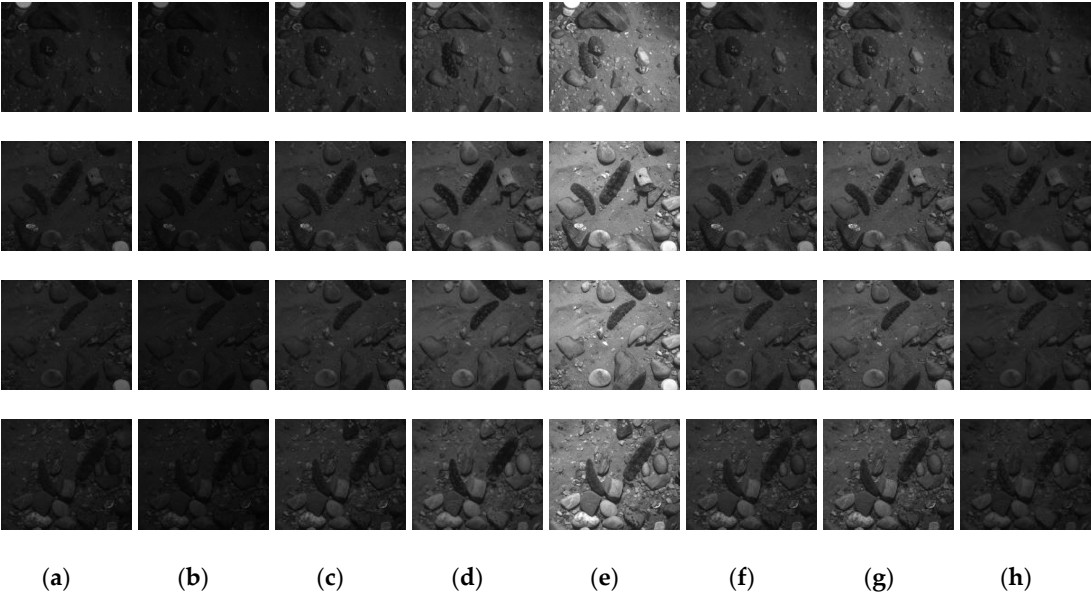

|  (a)  |  (b)  |  (c)  |  (d)  |  (e)  |  (f)  |  (g)  |  (h)  |

**Figure 10.** Image acquisition with different bands: (**a**) 470; (**b**) 490; (**c**) 510; (**d**) 540; (**e**) 570; (**f**) 650; (**g**) 740; (**h**) 820 nm.

CEM was used to detect the sea cucumbers in the composite image of each band subset. The detection results corresponding to each method are shown in Figure 11.

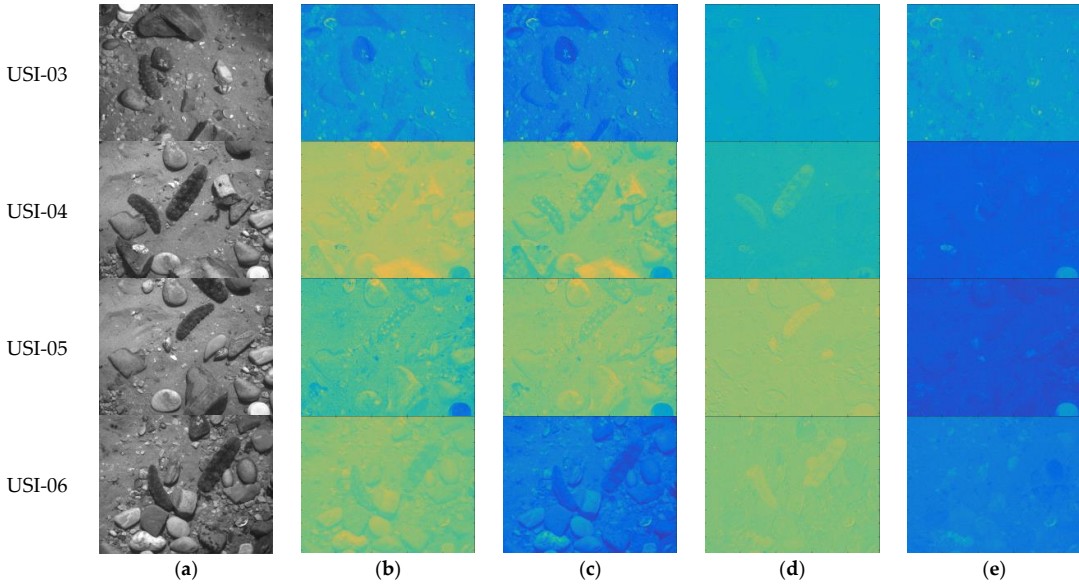

|  (a)  |  (b)  |  (c)  |  (d)  |  (e)  |

**Figure 11.** Detection map results using four BS (band selection) methods: (**a**) original image; (**b**) MinV-BP: minimum variance band priority; (**c**) MinV-BP-OIF: minimum variance band priority with OIF; (**d**) CTOIFBS: constrained-target optimal index factor (OIF) band selection; (**e**) UBS: uniform band selection.

The detection results shown in Figure 11 illustrated that when the set of bands selected by CTOIFBS was used to detect sea cucumbers, non-target pixels could be removed more effectively compared to other BS methods. On the contrary, MinV-BP and MinV-BP-OIF had poor ability in distinguishing the targets from the background, and the response to non-target pixels was also high when the target was detected. Table 12 shows the AUC values of the detection, and we also highlight the best and worst results by red and green. According to the AUC values of ($P_D$, $P_F$) in Table 12, UBS has the worst performance on all four test images. This shows that BS methods based on a constrained-target are more conducive to target detection. Furthermore, except for image USI-06, the AUC value of CTOIFBS is the highest. This proves that compared to other BS methods based on a constrained-target, MinV-BP, and MinV-BP-OIF, CTOIFBS has a better ability to characterize targets.

**Table 12.** AUC (area under the curve) values of detection using four BS (band selection) methods.

| Data | Method | USI-03 | USI-04 | USI-05 | USI-06 |
|------|--------|--------|--------|--------|--------|
| ($P_D$, $P_F$) | MinV-BP | 0.6343 | 0.6008 | 0.5335 | 0.7845 |
| | MinV-BP-OIF | 0.7061 | 0.7007 | 0.6414 | 0.8394 |
| | CTOIFBS | 0.8603 | 0.8500 | 0.7727 | 0.7859 |
| | UBS | 0.5997 | 0.5236 | 0.5915 | 0.5460 |
| ($P_D$, $\tau$) | MinV-BP | 0.8644 | 0.9074 | 0.8831 | 0.8657 |
| | MinV-BP -OIF | 0.8660 | 0.8316 | 0.9042 | 0.8893 |
| | CTOIFBS | 0.9123 | 0.9024 | 0.9177 | 0.9225 |
| | UBS | 0.9134 | 0.9611 | 0.9634 | 0.9609 |
| ($P_F$, $\tau$) | MinV-BP | 0.8349 | 0.8868 | 0.8715 | 0.7950 |
| | MinV-BP -OIF | 0.8262 | 0.7641 | 0.8707 | 0.8189 |
| | CTOIFBS | 0.9367 | 0.9614 | 0.9579 | 0.9606 |
| | UBS | 0.9244 | 0.9573 | 0.9528 | 0.9532 |

MinV-BP: minimum variance band priority; MinV-BP-OIF: minimum variance band priority with OIF; CTOIFBS: constrained-target optimal index factor (OIF) band selection; UBS: uniform band selection.

## 4. Conclusions

Hyperspectral imaging technology has advantages of high spectral resolution and abundant spectral information. Its applications to underwater object detection can help overcome the problems of a poor underwater imaging environment and complex background. The fast processing of detecting underwater hyperspectral targets can be achieved by CTOIFBS, while retaining crucial spectral information. In the meantime, CTOIFBS also overcomes the imaging and processing speed problems. Experiments show that the detection performance of the band subset selected by CTOIFBS is better than that by using other BS methods.

**Author Contributions:** Conceptualization, X.F. and M.S.; formal analysis, H.Y.; methodology, X.S. (Xiaodi Shang) and X.S. (Xudong Sun); writing—original draft, X.S. (Xiaodi Shang) and X.S. (Xudong Sun); writing—review and editing, C.-IC. All authors have read and agreed to the published version of the manuscript.

**Funding:** This research was funded by the National Nature Science Foundation of China, grant number 61601077, 61971082, 61890964; Fundamental Research Funds for the Central Universities, grant number 3132019341; State Administration of Foreign Experts Affairs, grant number ZD20180073.

**Conflicts of Interest:** The authors declare no conflict of interest.

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
