# Peer review of "Underwater Hyperspectral Target Detection with Band Selection"

_remotesensing, doi:10.3390/rs12071056_

Round 1

Reviewer 1 Report

This manuscript describes a relevant sequential application of different methods from the literature to improve underwater detection capability.
The objective is to arrive at a deployable operational system that is effective in terms of response time.
The manuscript is generally well written. The Authors have provided a well-structured exposition of their material. The problematic addressed is clearly introduced. It is developed and discussed correctly, making the whole manuscript pleasant to read.
The analysis provided and corresponding figures are quite appropriate to the text and its content. The list of references to the literature related to this field is also appropriate.
From my point of view, the whole content contributes to providing added-value to the reader.
However, I have noted in some places a lack of details and also some questions that may help to further improve the quality of the content.

1) The proof of concept is illustrated here for a quite limited number nBS=5. So, a first question relates to the maximum number of selected bands (nBS) beyond which the envisaged system is no longer advantageous, as it is too costly (particularly in terms of combinatorial calculation time). Does this number exist? In other words, where is the limit of applicability?

2) The same question arises in terms of material realization (due to the necessity to operate a spectral filter wheel).

3) If I understand the true interest of conducting a methodological comparison between different methods on a set of controlled data, the acquisition conditions in an underwater environment are much more constraining than in above water (fast attenuation). So, the comparison carried out on a non underwater hyperspectral image (real HYDICE Image) makes me question its relevance to the problem at hand. It would have been more convincing to show other experimental results in the underwater context.

4) This is a minor remark, but please define P_D, P_F and \tau as well as the way to set them appropriately.

5) In the comparison performed, the same number of bands is used for all methods. I think this should be discussed and justified, as each method may have an optimal operational point with a specific number of bands. Note that moreover, the spectral bands selected are quite different from one method to another one.

6) It would be interesting to show and analyse the intercorrelation between the complete set of spectral bands selected by the different BS methods.

7) Looking at figure 10, isn't the improvement observed mainly in the more efficient removal of non-target background pixels?

Typo(s):
Note it is preferable to generate and provide a file without the correction mode.
... have vert high band correlation ...
... where the best and worst results are highlighted by red and green respectively ... unfortunately, there's nothing highlighted in my file

Reviewer 2 Report

This manuscript aim  to use the under water hyperspectral image and BS technique to selection the suitable spectral bands in the selected application that importance and it is the good topic to do. However, this manuscript have some point could be improve before publish.

The Abstract: it should be complete in their own part. So, at the Line 27. What are the methods you have coppered to?

Table 2.Please provide the meaning of each parameter of AUC in the table (each column). It rather difficult to see the results details in the table form, may decide the use the other ways to present or it will be better if you can highlight as mentioned in the text.

The results show that full bands can provided the best results BUT I think full band have some problem such as the processing time, limited of real-tine application and some over-fitting in some model. it will be very good if authors can show some limited of full bands.

please provide the reason why authors select the sea cucumber as the sample. What is the difficulty or challenges to detection when compare to others.

The results form Table 4: it will be better if authors can overlay the spectral bands location (points) with the reflectance of sea cucumber and the background. So reader will learn more why the BS select that location and will make the very good discussion and improve the quality of the paper.

Line 316-319: I do not think you can say like this, so please explain more about the figure 6. sometime need be have the pointer to the sub-figure and please provide the reference figure (i.e., ground truth) as one of sub-figure.

The paper still lack of the discussion please provide more why your experimental give the results like this and your resuls related with some literatures and change the "Results" part to "Results and Discussion".

Reviewer 3 Report

This proposes a new band selection algorithm for hyperspectral images called Constrianed-target  optimal index factor band selection (CTOIFBS). Experimental results show that it can be superior than UBS, MinV-BF-OIF, and MinV-BF.

The difference between CTIOFBS and MinV-BF-OIF is not clear enough. (ie line 244 should be further explained). Topographical errors such as line 175 "vert" should be checked.
